# STORM: Spatio-Temporal Reconstruction Model for Large-Scale Outdoor Scenes

**Jiawei Yang**[⋆,¶]**, Jiahui Huang**[¶]**, Yuxiao Chen**[¶]**, Yan Wang**[¶]**, Boyi Li**[¶]**, Yurong You**[¶]**, Maximilian Igl**[¶]**, Apoorva Sharma**[¶]**,**
**Peter Karkus**[¶]**, Danfei Xu**[$,¶]**, Boris Ivanovic**[¶]**, Yue Wang**[†,⋆,¶]**, Marco Pavone**[†,§,¶]

[⋆] {yangjiaw,yue.w}@usc.edu, University of Southern California

[$] danfei@gatech.edu, Georgia Institute of Technology

[§] pavone@stanford.edu, Stanford University

[¶] {jiahuih,yuxiaoc,yanwan,boyil,yurongy,apoorvas,migl,
pkarkus,bivanovic}@nvidia.com, NVIDIA Research

[†] Equal advising.

## Abstract

We present STORM, a spatio-temporal reconstruction model designed for reconstructing dynamic outdoor scenes from sparse observations. Existing dynamic reconstruction methods often rely on per-scene optimization, dense observations across space and time, and strong motion supervision, resulting in lengthy optimization times, limited generalization to novel views or scenes, and degenerated quality caused by noisy pseudo-labels for dynamics. To address these challenges, STORM leverages a data-driven Transformer architecture that directly infers dynamic 3D scene representations—parameterized by 3D Gaussians and their velocities—in a single forward pass. Our key design is to aggregate 3D Gaussians from all frames using self-supervised scene flows, transforming them to the target timestep to enable complete (*i.e.*, "amodal") reconstructions from arbitrary viewpoints at any moment in time. As an emergent property, STORM automatically captures dynamic instances and generates high-quality masks using only reconstruction losses. Extensive experiments on public datasets show that STORM achieves precise dynamic scene reconstruction, surpassing state-of-the-art per-scene optimization methods (+4.3 to 6.6 PSNR) and existing feed-forward approaches (+2.1 to 4.7 PSNR) in dynamic regions. STORM reconstructs large-scale outdoor scenes in 200ms, supports real-time rendering, and outperforms competitors in scene flow estimation, improving 3D EPE by 0.422m and $Acc_5$ by 28.02%. Beyond reconstruction, we showcase four additional applications of our model, illustrating the potential of self-supervised learning for broader dynamic scene understanding. For more details, please visit our project page.

## 1 Introduction

Understanding and reconstructing dynamic 3D scenes from visual data is a fundamental challenge in computer vision, with significant applications in autonomous driving, robotics, and mixed reality, among many others. While static scene reconstruction methods have evolved from per-scene optimization (Mildenhall et al., 2021; Kerbl et al., 2023) to more data-driven approaches that leverage generalizable priors for improved performance (Zhang et al., 2024; Tang et al., 2024; Xu et al., 2024; Gao et al., 2024b; Wu et al., 2024b), most dynamic scene reconstruction methods still rely heavily on per-scene optimization, dense spatio-temporal observations (Park et al., 2021; Yang et al., 2024b), and strong motion supervision, such as dynamic objects' masks (Li et al., 2021b; Wang et al., 2024b), optical flow (Li et al., 2021b), or point trajectories (Wang et al., 2024b). Consequently, these models suffer from noise in the above pseudo-labels, require lengthy training times that range from hours to days, and cannot benefit from the data-driven advancements (*e.g.*, scaling laws (Zhai et al., 2022)) that are nowadays exploited by generalizable static reconstruction methods.

Our goal is to develop a scalable and data-driven solution for dynamic scene reconstruction addresses the limitations of existing methods. To this end, we present STORM, a self-supervised approach for reconstructing dynamic 3D scene representations and scene motions directly from sparse, multi-timestep, posed camera images. STORM leverages a Transformer model (Vaswani et al., 2017; Dosovitskiy, 2020) to reconstruct dynamic scenes in a *single* feed-forward pass, dramatically reducing reconstruction times from hours to seconds while harnessing data priors learned from large-scale datasets. Crucially, unlike existing methods that require pseudo-labels, STORM utilizes a self-supervised reconstruction loss, enabling significantly more cost-efficient data acquisition.

STORM is enabled by our proposed bottom-up amodal aggregation and transformation framework. Specifically, for each frame, we predict pixel-aligned or patch-aligned 3D Gaussian Splats (3DGS) (Kerbl et al., 2023) along with their motions, capturing the instantaneous state of the scene at each timestep. Since these image-aligned 3DGS can only represent the observed region from the context frames, we transform the Gaussians predicted from all the input frames to the target timestep, aggregating them into an "amodal" representation of the dynamic scene (Huang et al., 2022). By minimizing the reconstruction loss defined over this aggregated representation, our approach achieves accurate, self-supervised estimation of the temporal changes in the scene, as inaccuracies in dynamics would result in poor aggregation and transformation results, thereby introducing large reconstruction errors–driving our model toward better scene dynamics estimation.

Building upon this foundation, we introduce motion tokens—a set of learnable tokens prepended to the Transformer's input sequence. These motion tokens interact with image tokens through self-attention operations and are decoded as motion bases at the end of Transformer. They are designed to capture common motion primitives over time while also regularizing the degrees of freedom in predicted motions, motivated by the fact that the scene elements often move cohesively as groups (Wang et al., 2024b; Lei et al., 2024; Luiten et al., 2023). Concretely, we represent the motion of each 3D Gaussian using 3D velocity vectors, with the final motion computed as a weighted combination of shared velocity bases. These weights are determined by the similarity between motion tokens and image tokens. Consequently, motion tokens not only encode scene dynamics but also enable unsupervised segmentation of dynamic instance or motion group segmentation.

Lastly, we introduce a few practical techniques to make STORM more robust for in-the-wild captures. We address challenges such as sky modeling and camera exposure mismatches using auxiliary sky and affine tokens, and improve large novel view extrapolation and fine-grained human motions, such as leg and arm movements, using latent Gaussians and a latent decoder.

We conduct extensive experiments on the Waymo Open dataset (Sun et al., 2020), NuScenes (Caesar et al., 2020) and Argoverse2 (Wilson et al.) to evaluate the performance of STORM. The results demonstrate that STORM accurately reconstructs dynamic scenes in real-time (0.2 seconds for a 2-second clip), significantly surpassing per-scene optimization methods and other generalizable feed-forward models in terms of photorealism, geometry and motion estimation quality. These findings highlight the potential of self-supervised learning for advancing dynamic scene reconstruction and understanding. Our contributions can be summarized as follows:

- We propose STORM, the *first* feed-forward, self-supervised method for fast and accurate reconstruction of dynamic 3D scenes from sparse, multi-timestep, posed camera images.
- We propose a bottom-up framework that aggregates and transforms per-frame 3D Gaussian Splats into a cohesive scene representation, which enables self-supervised motion estimation. Furthermore, we introduce motion tokens that capture common motion primitives and regularize motion predictions, facilitating dynamic motion group segmentation without explicit motion or correspondence supervision.
- We present several enhancements for in-the-wild scenarios, including sky modeling, camera exposure inconsistency handling, large novel-view extrapolation, and fine-grained human motions reconstruction, making STORM well-suited for real-world applications.

## 2 RELATED WORK

**Dynamic scene reconstruction.** Derived from neural radiance fields (NeRFs) (Mildenhall et al., 2021), previous NeRF-based approaches model scene dynamics either by applying deformations to a canonical volume (Pumarola et al., 2021; Tretschk et al., 2021; Cao & Johnson, 2023; Park et al., 2021; Wu et al., 2022; Fridovich-Keil et al., 2023), or by chaining point-level scene mo-

tions (Xian et al., 2021; Gao et al., 2021; Li et al., 2021b; 2023; Liu et al., 2023). These per-scene optimization methods typically require dense temporal and spatial observations or explicit motion supervision, such as optical flow (Li et al., 2021b; 2023; Yang et al., 2023b; Gao et al., 2024a; Karaev et al., 2023; Fischer et al., 2024a) or dynamic masks (Liu et al., 2023; Li et al., 2023), to overcome the ill-posed nature of reconstructing dynamic scenes from sparse views. More recently, methods based on 3D Gaussian Splatting (3DGS) (Kerbl et al., 2023) have adopted similar strategies, applying deformations in canonical space (Wu et al., 2024a; Yang et al., 2024b) or rigid transformations to particles (Luiten et al., 2023; Wang et al., 2024b). However, they also depend on dense views or motion supervision, and remain limited to per-scene optimization, lacking the ability to leverage data priors, except for *e.g.* Ren et al. (2024), that only focuses on object-level reconstructions. We instead propose a feed-forward model trained on large-scale datasets, enabling outdoor dynamic scene reconstruction without per-scene optimization or explicit motion supervision.

**Feed-forward reconstruction.** Feed-forward approaches for 3D reconstruction and rendering aim to generalize across scenes by learning from large datasets. Early works on generalizable NeRFs focus on object-level (Chibane et al., 2021; Johari et al., 2022; Reizenstein et al., 2021; Yu et al., 2021) and scene-level reconstruction (Suhail et al., 2022; Wang et al., 2021; Du et al., 2023; Wang et al., 2024a). These methods typically rely on epipolar sampling or cost volumes to fuse multi-view features, requiring extensive point sampling for rendering, which results in slow speed and often unsatisfactory details. More recently, feed-forward models based on 3DGS have been proposed (Szymanowicz et al., 2024b; Charatan et al., 2024; Wewer et al., 2024; Zhang et al., 2024; Szymanowicz et al., 2024a; Tang et al., 2024; Xu et al., 2024). These models leverage large-scale object-centric synthetic datasets (Chang et al., 2015; Deitke et al., 2023; 2024) or indoor datasets (Zhou et al., 2018) for improved speed, view synthesis performance, and generalization. However, these methods are primarily designed for static scenes and struggle with dynamics. Unlike them, our approach is a 3DGS-based feed-forward model operated on large-scale outdoor *dynamic* scenes; more importantly, it also recovers scene motions without explicit motion supervision.

**Reconstruction for outdoor urban scenes.** Building photorealistic reconstructions of dynamic urban scenes from on-car logs is crucial for autonomous driving, as it enables closed-loop training and testing. Recent work has shifted focus from reconstructing static scenes (Guo et al., 2023) to dynamic ones. Most existing methods for dynamic urban scene reconstruction rely on box annotations to ensure controllability; however, these require expensive ground truth labels (Wu et al., 2023; Chen et al., 2024; Yang et al., 2023a; Tonderski et al., 2024; Fischer et al., 2024b; Zhou et al., 2024a; Ost et al., 2021; Williams et al., 2024; Fischer et al., 2024a) and often degrade in performance when using noisy pseudo-labels (Yan et al., 2024; Zhou et al., 2024b). Methods that do not rely on box annotations typically lack controllability over individual objects (Yang et al., 2024a; Chen et al., 2023; Huang et al., 2024). Furthermore, these approaches are per-scene-based, do not leverage data priors, and require lengthy training times, ranging from hours (Yan et al., 2024; Yang et al., 2024a) to days (Xie et al., 2023). In contrast, our method is a fast, scalable feed-forward model that reconstructs dynamic urban scenes purely through *self-supervision* in *seconds*. By differentiating between different instance groups *emerged* from our motion tokens, our approach enables better decomposition and controllability.

## 3 SELF-SUPERVISED SPATIAL-TEMPORAL RECONSTRUCTION MODEL

**Problem formulation.** Our goal is to recover spatiotemporal scene representations from a set of posed images. Specifically, given a set of images $\mathbf{I}_t^v \in \mathbb{R}^{H \times W \times 3}$, with height $H$ and width $W$, captured at multiple timesteps $t$ and optionally from multiple viewpoints $v$, along with their corresponding camera intrinsic and extrinsic parameters, we aim to reconstruct the underlying appearance, geometry, and dynamics of the scene over the observed duration. The core challenge arises from the transient and incomplete nature of the data: each point in the 4D space-time volume is typically observed only *once*, making it difficult to infer a comprehensive spatiotemporal representation.

### 3.1 STORM

To address the aforementioned challenges, we propose STORM, as illustrated in Fig. 1. We adopt a Lagrangian representation by modeling scene elements as a set of 3D Gaussians (3DGS) (Kerbl et al., 2023) that translate over time. Specifically, we begin by predicting 3DGS for each frame,

Figure 1: **STORM Overview.** From sparsely observed context frames, STORM reconstructs per-frame 3D Gaussian splats (3DGS) and predicts their scene flows using prepended learnable motion tokens and a dynamic mask decoder. The mask decoder computes weights for combining motion bases, derived from the motion tokens, to obtain scene flows. These predicted scene flows enable the aggregation and transformation of 3DGS over time, while the predicted weights support unsupervised motion group segmentation. The learning process is guided purely by reconstruction losses.

capturing the instantaneous state of the scene at each timestep. To model dynamics, we task the model with predicting the *velocity* of each Gaussian. Using these velocities, we transform the 3DGS from their observed context timesteps into any target timestep. This process aggregates the per-frame predictions into a cohesive *amodal* representation that remains consistent over time. Notably, our method trains solely with reconstruction losses, avoiding reliance on external motion supervision, significantly reducing data requirements. Below, we introduce our method in detail.

**Network and input.** STORM builds upon a standard Transformer model (Vaswani et al., 2017; Dosovitskiy, 2020), similar to Zhang et al. (2024). Following standard Vision Transformers (Dosovitskiy, 2020), we divide images into 2D non-overlapping patches. To incorporate 3D spatial information, we extend this patching process to the Plücker ray map (Plucker, 1865), which encodes the ray origins and directions corresponding to each pixel. These ray origins and directions are computed based on the camera's intrinsic and extrinsic parameters. The concatenated patches are then embedded through a linear patch embedding layer to obtain *image tokens*. Lastly, temporal information is infused via a time embedding layer (Peebles & Xie, 2023), which maps a frequency-encoded time vector into a time embedding. The resulting input to the Transformer is a 1D sequence of image tokens, formed by summing the image embeddings, ray embeddings, and time embeddings.

**Output.** The main output of our model is a set of pixel-aligned 3D Gaussians (Kerbl et al., 2023), each defined as $\mathbf{g} \equiv (\boldsymbol{\mu}, \mathbf{R}, \mathbf{s}, o, \mathbf{c})$, where $\boldsymbol{\mu} \in \mathbb{R}^3$ and $\mathbf{R} \in \mathbb{SO}(3)$ represent the center and orientation, $\mathbf{s} \in \mathbb{R}^3$ indicates the scale, $o \in \mathbb{R}^+$ denotes the opacity, and $\mathbf{c} \in \mathbb{R}^3$ corresponds to the color. The orientation is parameterized using a 4D quaternion. The centers of the 3D Gaussians are computed from ray origins and directions, which are pre-computed from camera parameters, along with the ray distance, by $\boldsymbol{\mu} = \text{ray}_o + d \cdot \text{ray}_{dir}$, where $d$ is the 1-channel ray distance predicted by the model. By default, our model predicts $\{\mathbf{G}_t^v \in \mathbb{R}^{(H \times W) \times 12}\}$ from the ViT feature map $\{\mathbf{F}_t^v \in \mathbb{R}^{(H//p \times W//p) \times e}\}$ using a linear layer, where $p$ is the patch size of the Transformer, and $e$ is the channel dimension. These 3D Gaussians exist independently in 3D space for each timestep. Next, we describe how their dynamics are obtained and how they are aggregated to form the final amodal, synchronized scene representations. For clarity, the view index $v$ is omitted unless necessary.

**Scene dynamics.** To capture the dynamics of a 3D scene, we model the motion of each 3D Gaussian using two velocity vectors, $\mathbf{v} \equiv [\mathbf{v}_t^-, \mathbf{v}_t^+] \in \mathbb{R}^6$ (we will detail how to compute these later), which represent the backward and forward velocities of a Gaussian at timestep $t$. Empirically, we find that assuming the Gaussians move with constant velocity within the duration of the clip (typically around 2 seconds) achieves a good balance between model complexity and representational power. Accordingly, the translation of a Gaussian at time $t'$ is specified as follows:

$$\boldsymbol{\mu}_{t \to t'} = \begin{cases} \boldsymbol{\mu}_t - (t' - t)\mathbf{v}_t^- & t' < t \\ \boldsymbol{\mu}_t + (t' - t)\mathbf{v}_t^+ & t' > t \end{cases}. \tag{1}$$

**Amodal aggregation.** To create a unified representation of the scene that is consistent over time, we aggregate the per-frame 3D Gaussians into an *amodal*, synchronized representation. Specifically, the Gaussians $\mathcal{G}_{t'}$ at an arbitrary target timestep $t'$ are defined as the union:

$$\mathcal{G}_{t'} = \bigcup_t \mathbf{G}_{t \to t'}, \tag{2}$$

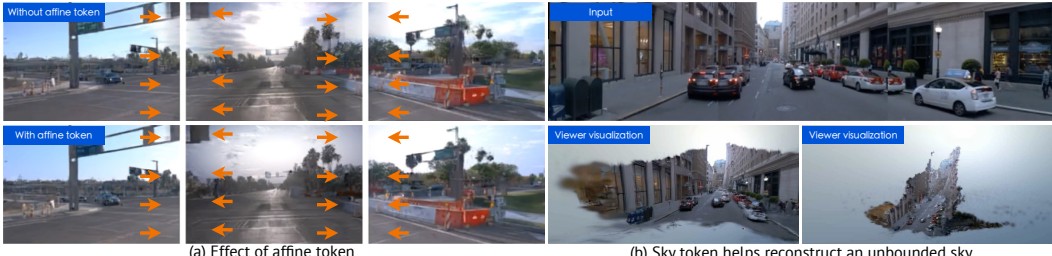

(a) Effect of affine token          (b) Sky token helps reconstruct an unbounded sky

Figure 2: **Effect of affine and sky tokens.** (a) The affine token handles exposure mismatches between cameras, eliminating artifacts like the black foggy floaters caused by exposure differences (orange arrows). (b) The sky token enables our method to predict sky colors for every pixel during rendering, even when they are not observed in any context frames.

where $\mathbf{G}_{t \to t'}$ contains translated Gaussians with centers $\boldsymbol{\mu}_{t \to t'}$ derived from the prediction $\mathbf{G}_t$. This amodal representation combines observations from multiple timesteps, capturing the complete geometry and appearance of the scene along with its dynamics. It supports tasks such as rendering the scene from novel viewpoints and moments in time.

**Motion tokens and mask decoder.** Motivated by the observation that scene dynamics often exhibit low-dimensional structures composed of shared motion patterns (Wang et al., 2024b; Kratimenos et al., 2023; Lei et al., 2024), we introduce $M$ learnable *motion tokens* (indexed by $m$), where $M \ll N$ and $N$ is the number of image tokens. These motion tokens are prepended to the input sequence of the Transformer and interact with other input tokens via self-attention. STORM leverages these tokens to capture common motion primitives present in the scene over time. At the output of the Transformer, the motion tokens are decoded into *velocity bases* $\mathbf{vb} \equiv (\mathbf{vb}^-, \mathbf{vb}^+) \in \mathbb{R}^{M \times 6}$ and *motion queries* $\mathbf{q} \in \mathbb{R}^{M \times e'}$ via a set of Multi-Layer Perceptrons (MLPs).[1] Here, $e'$ denotes the dimension of the motion embedding space. Simultaneously, the image embeddings $\mathbf{F} \in \mathbb{R}^{(H//p \times W//p) \times e}$ are mapped into this space to produce *pixel-aligned motion keys* $\mathbf{k} \in \mathbb{R}^{(H \times W) \times e'}$ through several deconvolution layers, where each key vector $\mathbf{k}_{i,j} \in \mathbb{R}^{e'}$ corresponds to a spatial location $(i, j)$.

Inspired by SAM (Kirillov et al., 2023), we compute the similarity between motion queries $\mathbf{q}$ and motion keys $\mathbf{k}$ to derive weights $\mathbf{w} \in \mathbb{R}+^{(H \times W) \times M}$ for combining the velocity bases. Specifically, the weights $\mathbf{w}^{(i,j)} \in \mathbb{R}+^{M}$ at each spatial location $(i, j)$ are computed as:

$$w_m^{(i,j)} = \frac{\exp\left(\frac{\mathbf{q}_m \cdot \mathbf{k}_{i,j}}{\tau}\right)}{\sum_{m'=1}^{M} \exp\left(\frac{\mathbf{q}_{m'} \cdot \mathbf{k}_{i,j}}{\tau}\right)}, \tag{3}$$

where $\tau$ is a temperature hyperparameter that controls the sharpness of the distribution (we set $\tau = 0.5$ in all experiments). The weights $\mathbf{w}$ are then used to combine the velocity bases for each Gaussian associated with the pixel at location $(i, j)$, yielding the final velocity $\mathbf{v}$ used in Eq. (1):

$$\mathbf{v}^{(i,j)} = \sum_{m=1}^{M} w_m^{(i,j)} \mathbf{vb}_m, \quad \text{where} \sum_{m=1}^{M} w_m^{(i,j)} = 1 \text{ and } 0 \leq w_m^{(i,j)} \leq 1. \tag{4}$$

This design captures the low-dimensional structure of scene dynamics and regularizes the motion prediction problem by reducing its degrees of freedom.

## 3.2 STORM IN THE WILD

Modeling unbounded scenes from multi-view videos captured in the wild introduces additional challenges, such as representing the sky, handling exposure differences between cameras, and accurately modeling humans.

**Auxiliary tokens for sky and exposure mismatches.** In in-the-wild video collections, particularly those captured by autonomous vehicles, sky modeling and exposure mismatches are common challenges. Specifically, the sky often lacks well-defined depth, and the same 3D point may appear with

---

[1]Following the mask decoder design of SAM (Kirillov et al., 2023), we use distinct MLP weights for each motion token, as this leads to cleaner motion masks.

Time

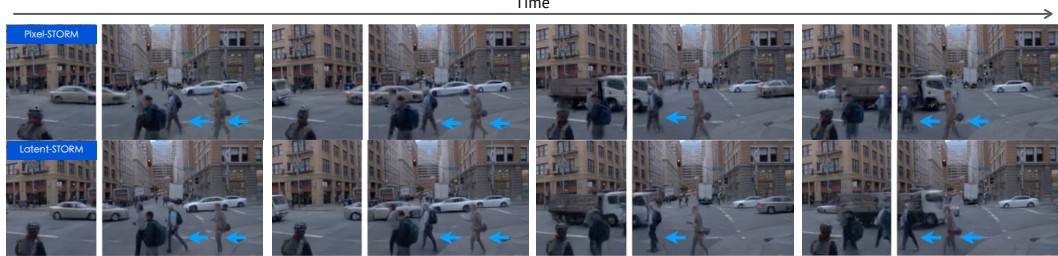

Figure 3: **Latent-STORM Examples.** Using latent Gaussians and a decoder, Latent-STORM can photorealistically reconstruct human leg movements. Note how the leg angles change over time in Latent-STORM, while they remain static in regular STORM (blue arrows). Please refer to "Human Modeling with Latent-STORM" section in our project page for video comparisons.

varying colors across different images due to exposure differences between cameras. To address these issues, we introduce two types of learnable auxiliary tokens into the input sequence: the *sky token* and the *affine token*, designed similarly to the motion tokens.

A single sky token is used to capture sky information. At the output of the Transformer, this sky token conditions a modulated MLP that takes the ray direction $\mathbf{d}$ as input and outputs the sky color:

$$\mathbf{c}_{\text{sky}} = \text{MLP}_{\text{sky}}\left(\gamma\left(\mathbf{d}\right); \text{sky\_token}\right), \tag{5}$$

where $\gamma(\cdot)$ is a frequency-based positional embedding function as in Mildenhall et al. (2021). This setup allows us to query sky colors for every pixel we wish to render. Given a rendered image $\mathbf{I}_{\text{GS}}$ before sky composition and a rendered opacity map $\hat{\mathbf{O}}$, the final image with the sky composed is:

$$\mathbf{I}' = \mathbf{I}_{\text{GS}} + (1 - \hat{\mathbf{O}}) \cdot \mathbf{c}_{\text{sky}}. \tag{6}$$

To handle exposure mismatches between cameras, we introduce $v$ learnable affine tokens, where $v$ is the number of cameras. These tokens aim to capture exposure variations between cameras. At the Transformer's output, each affine token is mapped to a scaling matrix $\mathbf{S} \in \mathbb{R}^{3 \times 3}$ and a bias vector $\mathbf{b} \in \mathbb{R}^3$ via a linear layer. The final rendered image is obtained by applying the affine transformation to every pixel: $\hat{\mathbf{I}} = \mathbf{S}\mathbf{I}' + \mathbf{b}$. The affine transformation has been similarly explored in previous work (Rematas et al., 2022) but only in a per-scene optimization setting. Examples illustrating the roles of the sky and affine tokens are shown in Fig. 2.

**Latent-STORM.** As an *optional* enhancement, we introduce the use of latent Gaussians coupled with a latent decoder to improve STORM's performance on large novel view extrapolation and human body modeling. Instead of predicting pixel-aligned Gaussians with a 3-channel color vector, we predict *patch-aligned* Gaussians with a $c$-channel latent vector. This approach reduces the number of Gaussians while increasing the modeling capacity of each Gaussian. Consequently, the model output changes from $\mathbf{G}_t^v \in \mathbb{R}^{(H \times W) \times 12}$ to $\mathbf{G}_t^v \in \mathbb{R}^{(H//p \times W//p) \times (9+c)}$, where $p$ is the patch size of the Transformer. After rasterization, we obtain a $p\times$ downsampled latent feature map $\mathbf{F}$, which is composited with a learnable *inpainting token* using the opacity map: $\hat{\mathbf{F}} = \mathbf{F} + (1 - \hat{\mathbf{O}}) \cdot \text{inpainting\_token}$. This composited feature map is then upsampled to the original resolution using a convolutional decoder (Rombach et al., 2022) to produce the final color and depth outputs.

This design addresses the limitations of color-based Gaussians in handling occluded regions that are not visible in any of the context views, since the decoder can infer and reconstruct these unseen areas from the inpainting tokens within a reasonable extrapolation range.[2] Additionally, capturing fine-grained human motions, such as leg and arm movements, remains challenging with sparse observations. However, we find that the decoder can photorealistically recover these subtle motions, albeit with a slight compromise in pixel sharpness due to the additional decoding process. Fig. 3 compares this property. We refer to this new variant of our method as Latent-STORM.

### 3.3 IMPLEMENTATION

**Model architecture.** By default, we use a 12-layer Vision Transformer (ViT-B) (Dosovitskiy, 2020) with full attention and a patch size of 8, along with $M = 16$ motion tokens. We study how perfor-

---

[2]Significant hallucination would require stronger generative capabilities, which we leave as future work.

mance scales with respect to the number of motion tokens in Appendix B.2. For the mask decoder and MLP components, we adopt the implementation from SAM (Kirillov et al., 2023), and set the final projected motion embedding space to be 32 dimensional. For the modulated MLP used in sky modeling, we follow DiT (Peebles & Xie, 2023) to use an adaptive `LayerNorm` layer for modulation. Our GS backend is based on `gsplat` (Ye et al., 2024).

**Supervision and loss functions.** After aggregating the amodal scene representation from all observed timesteps, we transform it into the target timesteps we wish to render. During training, we randomly select a starting timestep $t$ and sample 4 target timesteps $t'$ within the range $[t, t + 2s]$ for supervision. Using the transformed amodal Gaussians $\mathcal{G}_{t'}$, we minimize a combination of reconstruction loss, sky loss, and velocity regularization loss:

$$\mathcal{L} = \mathcal{L}_{\text{recon}} + \lambda_{\text{sky}} \cdot \mathcal{L}_{\text{sky}} + \lambda_{\text{reg}} \cdot \mathcal{L}_{\text{reg}}, \tag{7}$$

where the reconstruction loss $\mathcal{L}_{\text{recon}}$ includes RGB loss, perceptual loss, and depth loss. The sky loss $\mathcal{L}_{\text{sky}}$ encourages zero opacity for Gaussians located in the sky-region. The velocity regularization loss is defined as $\mathcal{L}_{\text{reg}} = \|\mathbf{v}\|_2^2/3$, where we encourage the predicted velocity vectors to be small. $\lambda_{\text{sky}} = 0.1$ and $\lambda_{\text{reg}} = 0.005$ are two hyperparameters that balance different losses. Details regarding training, implementation, and the loss functions are provided in Appendix A.

## 4 EXPERIMENTS

**Datasets.** We primarily conduct experiments on the Waymo Open Dataset (Sun et al., 2020), which contains 1,000 sequences of driving logs: 798 sequences for training and 202 for validation. Each sequence consists of a 20-second video recorded at 10FPS. For training and testing, we use the frontal three cameras at an $8\times$ downsampled resolution $(160 \times 240)$. The input to our model consists of 4 context timesteps, evenly spaced at $t + 0s$, $t + 0.5s$, $t + 1.0s$, and $t + 1.5s$, where $t$ is a randomly chosen starting timestep. Additionally, we evaluate our method on the NuScenes (Caesar et al., 2020) and Argoverse2 (Wilson et al., 2023) datasets. Please refer to Appendix B.1 for more details.

### 4.1 RENDERING

**Setup.** We assess novel view synthesis from sparse view reconstructions using the validation set of the Waymo Open Dataset (Sun et al., 2020). Each video sequence is segmented into 10 non-overlapping clips, each 2.0 seconds long and consisting of 20 frames (3 camera views per frame). For reconstruction, we provide the 1st, 5th, 10th, and 15th frames as context frames, and evaluate on the remaining frames. This setup enables evaluation of both interpolation (0s to 1.5s) and extrapolation (1.5s to 2.0s), resulting in 2,019 video clips, or 96,912 total images. We report standard metrics: PSNR, SSIM, and Depth RMSE. Additionally, we analyze performance on both full images and dynamic regions for a more comprehensive evaluation. Lastly, we report the inference time; for per-scene optimization methods, this refers to the test-time fitting time, while for generalizable methods, it refers to the time required for the model to feedforward and output 3D Gaussians.

**Baselines.** We compare our method against two categories of approaches: per-scene optimization methods and feed-forward models. For per-scene optimization, we evaluate against a NeRF-based approach, EmerNeRF (Yang et al., 2024a), and 3DGS-based approaches, including 3DGS (Kerbl et al., 2023), PVG (Chen et al., 2023), and DeformableGS (Yang et al., 2024b). Since LiDAR data is not provided at *test* time in our setup, we train these baselines without LiDAR supervision to ensure a fair comparison. In the second category, we compare against recent large reconstruction models, including LGM (Tang et al., 2024) and GS-LRM (Zhang et al., 2024). We notice that the default LGM, which predicts raw 3DGS coordinates, performs poorly in our datasets. Therefore, we modify it to predict depth and recover positions similar to our approach, and denote it as LGM*. We provide more implementation details of these baselines in Appendix A.3.

**Results.** We present the quantitative results in Table 1. Compared to per-scene optimization methods, STORM achieves significantly better performance in both dynamic regions and full images in terms of photorealism, geometry, and inference speed. Specifically, in dynamic regions, STORM outperforms the best per-scene method by a substantial 5dB in PSNR and 0.346 SSIM. For full images, STORM attains around 0.5 to 1dB PSNR gain. Notably, STORM achieves these improvements while reducing inference time from tens of minutes to just 0.18 second, making it suitable for real-time

Table 1: **Comparison to state-of-the-art methods on the Waymo Open Dataset.** We compare photorealism, geometry and speed metrics against both per-scene optimization methods and generalizable feed-forward methods. PSNR, SSIM, and Depth RMSE (D-RMSE) are reported. Speed metrics are estimated on a single A100 GPU. *: reproduced by us. †: Non-sky region.

| Methods | Dynamic-only | | | Full image† | | | Inference speed | Real-time rendering |
|---|---|---|---|---|---|---|---|---|
| | PSNR↑ | SSIM↑ | D-RMSE↓ | PSNR↑ | SSIM↑ | D-RMSE↓ | Time↓ | (>200FPS) |
| *Per-Scene Optimization methods* | | | | | | | | |
| EmerNeRF (Yang et al., 2024a) | 17.79 | 0.255 | 40.88 | 24.51 | 0.738 | 33.99 | 14min | × |
| 3DGS (Kerbl et al., 2023) | 17.13 | 0.267 | 13.88 | 25.13 | 0.741 | 19.68 | 23min | ✓ |
| PVG (Chen et al., 2023) | 15.51 | 0.128 | 15.91 | 22.38 | 0.661 | 13.01 | 27min | ✓ |
| DeformableGS (Yang et al., 2024b) | 17.10 | 0.266 | 12.14 | 25.29 | 0.761 | 14.79 | 29min | ✓ |
| *Generalizable feed-forward methods* | | | | | | | | |
| LGM (Tang et al., 2024) | 17.36 | 0.216 | 11.09 | 18.53 | 0.447 | 9.07 | 0.06s | ✓ |
| LGM* (Tang et al., 2024) | 19.58 | 0.443 | 9.43 | 23.59 | 0.691 | 8.02 | 0.06s | ✓ |
| GS-LRM* (Zhang et al., 2024) | 20.02 | 0.520 | 9.95 | 25.18 | 0.753 | 7.94 | **0.02s** | ✓ |
| *Ours* | | | | | | | | |
| Latent-STORM | 21.26 | 0.535 | 9.42 | 25.03 | 0.750 | 8.57 | 0.18s | ✓ |
| STORM | **22.10** | **0.624** | **7.50** | **26.38** | **0.794** | **5.48** | 0.18s | ✓ |

Table 2: **Comparison to state-of-the-art methods on more datasets.** We report full-image PSNR and Depth RMSE metrics on the NuScenes (Caesar et al., 2020) and Argoverse2 (Wilson et al.) datasets.

| Method | NuScenes | | Argoverse2 | |
|---|---|---|---|---|
| | PSNR↑ | D-RMSE↓ | PSNR↑ | D-RMSE↓ |
| LGM | 23.21 | 7.34 | 22.93 | 14.20 |
| GS-LRM | 24.53 | 7.71 | 24.49 | 14.70 |
| Ours | **24.90** | **5.43** | **24.80** | **13.51** |

Table 3: **Comparison of scene flow estimation on the Waymo Open Dataset.** All competing methods require LiDAR input at test time, whereas our method relies solely on camera images.

| Methods | EPE3D $(m)$ ↓ | $Acc_5(\%)$ ↑ | $Acc_{10}(\%)$ ↑ | $\theta$ (rad) ↓ | Inference Time ↓ |
|---|---|---|---|---|---|
| NSFP (Li et al., 2021a) | 0.698 | 42.17 | 54.26 | 0.919 | ~27s/frame |
| NSFP++ (Najibi et al., 2022) | 0.711 | 53.10 | 63.02 | 0.989 | ~167s/frame |
| Ours | **0.276** | **81.12** | **85.61** | **0.658** | ~0.025s/frame |

applications. This confirms our motivation of building data-driven models that excels with data priors, addressing the limitations of per-scene optimization methods. Compared to other generalizable feed-forward models, STORM demonstrates a robust ability to model scene dynamics and process multi-timestep, multi-view images holistically. This enables us to model dynamic scenes better.

**Results on additional datasets.** We evaluate the applicability of STORM on the NuScenes (Caesar et al., 2020) and Argoverse2 (Wilson et al.) datasets, comparing against other generalizable methods in Table 2. We provide detailed setups in Appendix B.1. Our method achieves the best performance in both *full*-image PSNR and Depth RMSE metrics. Measuring performance on dynamic regions is expected to have more gains. These results validate the generalizability of STORM across datasets.

## 4.2 FLOW ESTIMATION

**Setup and baselines.** A unique capability of STORM is scene motion estimation, which we demonstrate using the Waymo Open Dataset (Sun et al., 2020). This dataset provides ground truth 3D scene flows, which we *do not* use for supervision. We measure 3D scene flow estimation accuracy using standard metrics following Li et al. (2021a): End-Point Error in 3D (EPE3D), $Acc_5$, $Acc_{10}$, angular error $\theta_{err}$, and inference time. For baselines, we compare STORM against NSFP (Li et al., 2021a), and NSFP++ (Najibi et al., 2022). Since existing methods cannot directly synthesize scene flows at novel timesteps, we evaluate the scene flows estimated on the context frames. Specifically, we provide the 1st, 5th, 10th, and 15th sensor observations as input and evaluate on these frames rather than on the remaining ones. Notably, all these competing methods require LiDAR input at *test time*, whereas STORM relies *solely* on camera images, making our comparisons highly conservative.

**Results.** As presented in Table 3, STORM consistently outperforms all methods across all metrics, achieving substantial improvements in EPE3D, $Acc_5$ and $Acc_{10}$ despite using only camera images as input. While NSFP (Li et al., 2021a) and NSFP++ (Najibi et al., 2022) excel at scene flow estimation with dense space-time observations (10Hz), they struggle with sparse observations (2Hz). In contrast, STORM demonstrates robust performance even with sparse data. To the best of our knowledge, STORM is the first scene flow estimation method that does not require depth signals at test time. These results demonstrate the effectiveness of our approach in explicit motion understanding and its potential for scene flow estimation without reliance on additional sensors at test time.

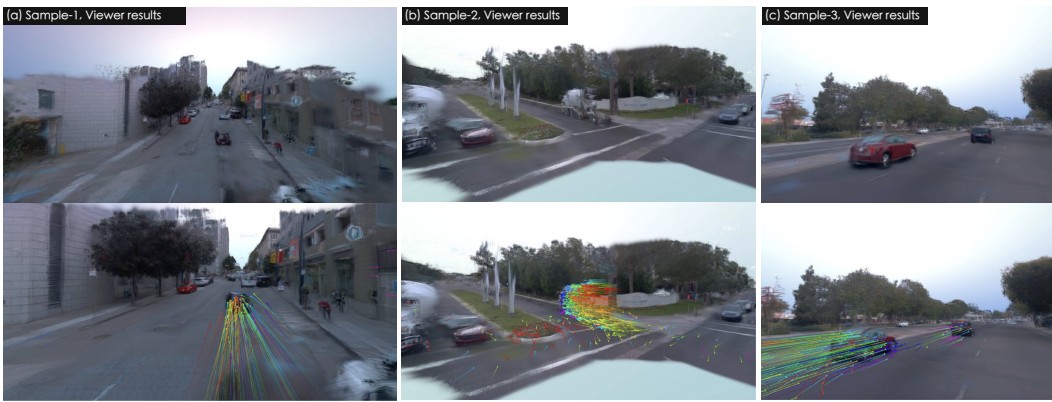

Figure 4: **Iterative reconstruction of static scenes.** STORM reconstructs 20-second-long videos within 1 second in an iterative manner, which can serve as initialization for per-scene optimization methods for further refinement.

Figure 5: **Iterative reconstruction of dynamic scenes. Top**: STORM reconstructs 20-second-long videos within 1 second in an iterative manner. **Bottom**: Furthermore, by chaining scene flows, we obtain point trajectories for dynamic Gaussians.

### 4.3 ABLATION STUDY, QUALITATIVE RESULTS AND APPLICATION

**Ablation study.** We present a detailed ablation study to analyze the impact of the velocity regularization coefficient $\lambda_{\text{reg}}$, the number of motion tokens $M$, and the number of input timesteps during training and testing in Appendix B.2. In short, without velocity regularization, training collapses because of gradient explosion, and an optimal $\lambda_{\text{reg}}$ (5$e$-3) yields the best performance. STORM is robust to the choice of $M$ and performs best with $M = 16$ for both rendering and flow estimation tasks. Furthermore, when trained on a fixed number of timesteps (*e.g.*, 4), STORM demonstrates strong zero-shot generalization to varying input timesteps (*e.g.*, 1, 2, 6, 10) at test-time, though re-training for specific configurations achieves optimal results.

**Larger scene reconstruction.** Figs. 4 and 5 show the results of applying STORM iteratively to 20-second posed videos, each includes 600 images captured across 3 cameras over 200 timesteps. We process videos clip by clip, completing the inference for an entire video within **0.5 seconds** on a single A100 GPU using batch inference and a bf16 precision. By merging the Gaussians predicted from each clip, we construct a comprehensive dynamic 3D scene reconstruction in a fully feedforward manner. Although some artifacts are present in overlapping regions when merging clips, these results demonstrate STORM's potential for holistic dynamic scene reconstruction, even for long and complex sequences. Furthermore, the predicted 3D Gaussians can serve as an initialization for per-scene optimization methods for further refinement, which we leave for future work.

**Point tracking.** While per-point trajectory estimation is not the primary focus of our work, STORM models the motion of Gaussians over time, enabling us to derive point trajectories by chaining scene flows. In the bottom row of Fig. 5, we present examples of point tracking, demonstrating STORM's potential for applications such as motion analysis and object tracking. We hope this approach could inspire further exploration in related tasks, such as 2D pixel or 3D point tracking.

**Scene flow estimation and motion segmentation.** We visualize the predicted 3D velocities and motion token assignments, *i.e.*, the motion segmentation mask decoded by the mask decoder, in Fig. 6. These masks are derived by applying an argmax operation on per-pixel assignment weights **w** along the motion token dimension $M$ (see Eq. (4)). These visualizations demonstrate that STORM captures scene dynamics and groups 3D Gaussians that correspond to the same moving pattern, resulting in motion-level segmentations. These unsupervised segmentations allow us to select Gaussians based on their assignments for editing, without using ground truth 3D bounding boxes.

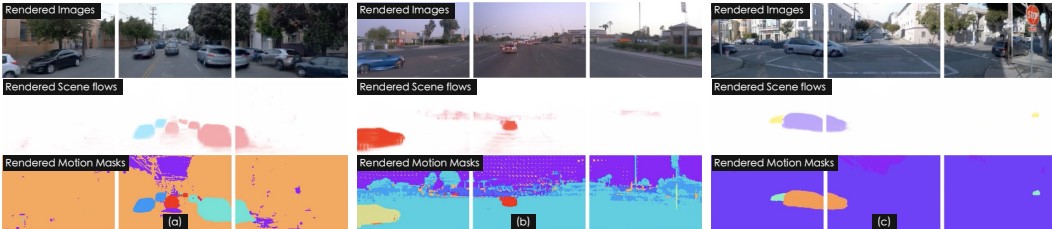

Figure 6: **Self-supervised scene flow estimation and motion segmentation.** For each sample, we show the rendered camera images (top), scene flows (middle), and motion assignments (bottom).

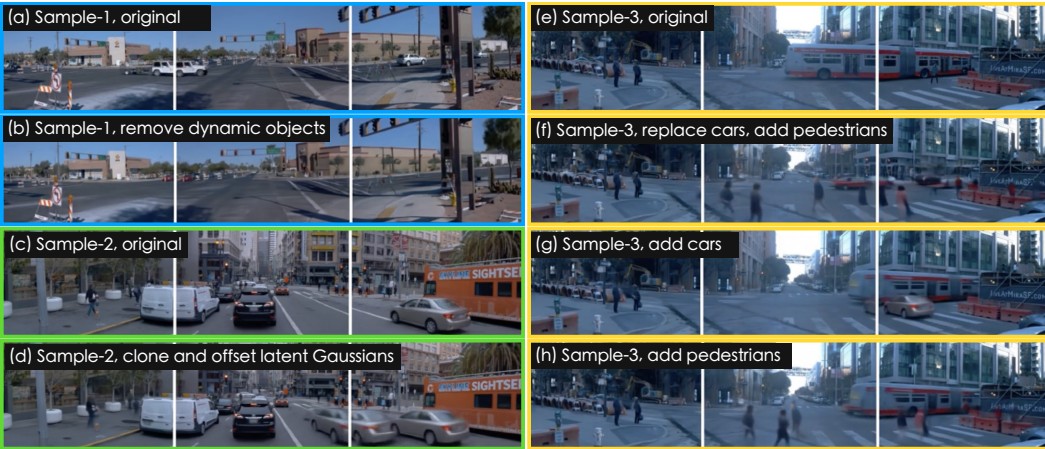

Figure 7: **STORM editing examples.** We present examples of removing or cloning vehicles (a-d), as well as adding or replacing pedestrians and vehicles (e-h). Notice how STORM harmonizes the edited images due to the use of the decoder. More examples can be found at our project page.

**Editing, control, and inpainting.** We demonstrate the capabilities of Latent-STORM[3] in editing scenes. Since each latent Gaussian still represents a physical particle in space, we can edit the scene by adding, removing, or modifying these Gaussians *before feature map rasterization and decoding*. As shown in Fig. 7, Latent-STORM can reconstruct human movements (f, h), hallucinate occluded regions, and provide other advantages, such as image harmonization (f, g, h). For instance, Latent-STORM can recover leg movements and hand gestures. Furthermore, one common challenge in driving scene reconstruction is the difficulty of inpainting occluded regions or removing static objects without leaving black holes. Latent-STORM addresses this by synthesizing these areas, albeit with slight blurriness. Please refer to our project page for more visualizations if interested. Overall, these results demonstrate the flexibility of STORM as a tool for scene editing and simulation.

## 5 CONCLUSION

In this work, we have introduced STORM, a scalable spatio-temporal model designed for dynamic scene reconstruction from sparse observations without requiring explicit motion supervision. Through extensive experiments, we have demonstrated STORM's ability to reconstruct dynamic outdoor scenes and estimate scene dynamics. Our method significantly surpasses existing per-scene optimization and feed-forward approaches, showcasing its versatility for a wide range of applications, including view synthesis, scene editing and point tracking. Looking ahead, we hope STORM will become a foundational model for various tasks across multiple domains, enabling more efficient and flexible approaches to 4D scene reconstruction, motion estimation, and beyond. As research in spatio-temporal modeling progresses, we believe STORM has the potential to unlock new possibilities for real-time dynamic scene analysis, interactive applications, and further advancements in self-supervised learning.

---

[3]Our default STORM performs well for vehicle editing but exhibits slightly reduced performance for human modeling.

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

# A IMPLEMENTATION DETAILS

In this section, we discuss the implementation details of STORM.

## A.1 STORM IMPLEMENTATION DETAILS

**Gaussian Parameterization.** Our Transformer architecture and output mapping largely follow Zhang et al. (2024). The input images are normalized to the range of [-1, 1]. Recall that each Gaussian is defined as $\mathbf{g} \equiv (\boldsymbol{\mu}, \mathbf{R}, \mathbf{s}, o, \mathbf{c})$, where $\boldsymbol{\mu} \in \mathbb{R}^3$ and $\mathbf{R} \in \mathbb{SO}(3)$ represent the center and orientation, $\mathbf{s} \in \mathbb{R}^3$ indicates the scale, $o \in \mathbb{R}^+$ denotes the opacity, and $\mathbf{c} \in \mathbb{R}^3$ corresponds to the color. Below, we describe the activations or normalizations applied to the raw outputs to derive these parameters.

For coordinates $\boldsymbol{\mu}$, we first compute ray origins and directions from the camera's intrinsic and extrinsic parameters. We then compute $\boldsymbol{\mu} = \text{ray}_o + d \cdot \text{ray}_{dir}$, where $d$ is the 1-channel ray distance predicted by our model. Depth $d$ is computed as $d = \text{near} + \sigma(d) * (\text{far} - \text{near})$, where $\sigma$ represents the sigmoid function, and near and far are hyperparameters. In all our experiments, we set near to 0.1 and far to 400.0.

For rotation $\mathbf{R}$, we parameterize it with 4-dimensional quaternion vectors. Note that there is a default normalization step in gsplat (Ye et al., 2024) that applies $L_2$ normalization to ensure quaternion vectors are unit vectors.

For scale $\mathbf{s}$, we compute $\mathbf{s} = \min(\exp(\mathbf{s}' - 2.3), 0.5)$, following Zhang et al. (2024), where $\mathbf{s}'$ represents the outputs before normalization. This regularization limits the maximum size of Gaussians and improves training stability.

For opacity $o$, we compute $o = \sigma(o' - 2.0)$, again following Zhang et al. (2024).

For color $\mathbf{c}$, we compute $\mathbf{c} = \sigma(\mathbf{c}') \cdot 2 - 1$. When computing PSNR, we map color back to [0, 1].

**Sky MLP.** The modulated sky MLP predicts sky color from view directions by conditioning a sky token $\mathbf{c}_{sky}$ through a modulated linear layer. Specifically, frequency-embedded viewing directional vectors $\gamma(\mathbf{d})$ are linearly projected to 64 dimensions, then normalized using LayerNorm without affine parameters. The sky token outputted from the Transformer serves as the conditioning vector $\mathbf{c}_{sky}$ (768 dims) is mapped to 64 dimensions and used in an adaptive layer normalization (AdaLN) process, where $\mathbf{c}_{sky}$ is transformed to produce shift and scale vectors, each of size 64, modulating the normalized features by $\mathbf{x} = \mathbf{x} \cdot (1 + \text{scale}) + \text{shift}$. Finally, the modulated features are linearly transformed to an output of 3-dimensional color.

**Mask Decoder.** The convolution layers in the mask decoder are similar to those used in SAM (Kirillov et al., 2023), which is defined as:

```
self.output_upscaling = nn.Sequential(
    nn.ConvTranspose2d(embed_dim, 512, kernel_size=2, stride=2),
    LayerNorm2d(512),
    nn.GELU(),
    nn.ConvTranspose2d(512, 256, kernel_size=2, stride=2),
    LayerNorm2d(256),
    nn.GELU(),
    nn.ConvTranspose2d(256, 128, kernel_size=2, stride=2),
    nn.GELU()
)
```

The input to this decoder is the ViT output feature maps. After they are upsampled by the mask decoder, they are projected into a 32-dimensional space using a linear layer. Additionally, motion tokens are mapped into a 32-dimensional space using a set of three-layer MLPs, following the design in SAM (Kirillov et al., 2023).

**Training.** We train our model for 100,000 iterations with a global batch size of 64 on NVIDIA A100 GPUs, using a learning rate of $4 \times 10^{-4}$. The training process utilizes the AdamW optimizer (Loshchilov & Hutter, 2019) along with a cosine learning rate scheduler that includes a linear warmup phase over the first 5,000 iterations. We enable the LPIPS loss (Zhang et al., 2018) only

after 5,000 iterations, as we find this approach stabilizes training. Gradient checkpointing is enabled by default to reduce memory usage. Behind the scene, we observe that STORM benefits from longer training durations and larger model sizes. We maintain the default setup to ensure alignment with our baseline in this work. However, an attractive direction for future work is to explore the scaling laws of STORM (Zhai et al., 2022).

**Point trajectory estimation.** We visualize the trajectories of dynamic Gaussians. These trajectories are obtained by chaining per-frame scene flows. Specifically, for each frame at $t$, we use the predicted scene flow to transform Gaussians to its next frame $t + 1$ to obtain its estimated destination. Then, for every Gaussian at $t + 1$, we find its nearest Gaussians transformed from $t$ and connect them to visualize the trajectories. This process is recursively applied to all frames to obtain the final trajectories.

## A.2 LOSS FUNCTION

We present more details about our loss function here. Given the rendered images $\hat{\mathbf{I}}$, depth maps $\hat{\mathbf{D}}$, opacity maps $\hat{\mathbf{O}}$, and velocities for all 3D Gaussians $\mathbf{v}$, along with the corresponding observed camera images $\mathbf{I}$, depth maps $\mathbf{D}$, and sky masks $\mathbf{M}$ that are predicted by a pre-trained segmentation model (Chen et al., 2022), we compute the overall loss as:

$$\mathcal{L} = \mathcal{L}_{\text{recon}} + \lambda_{\text{sky}} \cdot \mathcal{L}_{\text{sky}} + \lambda_{\text{reg}} \cdot \mathcal{L}_{\text{reg}}, \tag{A1}$$

where the reconstruction loss combines RGB loss, depth loss, and LPIPS loss (Zhang et al., 2018):

$$\mathcal{L}_{\text{recon}} = \|\hat{\mathbf{I}} - \mathbf{I}\|_2 + \|(\hat{\mathbf{D}} - \mathbf{D})/\max(\mathbf{D})\|_1 + \lambda_{\text{lpips}} \cdot \text{LPIPS}\left(\hat{\mathbf{I}}, \mathbf{I}\right), \tag{A2}$$

and the sky loss and velocity regularization loss are MSE losses that encourage sparsity:

$$\mathcal{L}_{\text{sky}} = \|\hat{\mathbf{O}} - (\mathbf{1} - \mathbf{M})\|_1, \quad \mathcal{L}_{\text{reg}} = \|\mathbf{v}\|_2^2/3 = \frac{1}{3}\sum_{i=1}^{3}\mathbf{v}_i^2. \tag{A3}$$

Here, the $\lambda$ terms control the relative weighting of each loss component. For the LPIPS loss, we utilize a VGG-19-based (Simonyan & Zisserman, 2014) implementation. We set $\lambda_{\text{lpips}}$ to $0.05$, $\lambda_{\text{sky}}$ to $0.1$, and $\lambda_{\text{reg}}$ to $5e\text{-}3$ in all experiments.

## A.3 BASELINE IMPLEMENTATIONS

For per-scene optimization 3DGS-based methods, we use the recently open-sourced codebase DriveStudio from Chen et al. (2024), which includes implementations for PVG (Chen et al., 2023), DeformableGS (Yang et al., 2024b), and 3DGS (Kerbl et al., 2023) on the Waymo Open Dataset. For EmerNeRF (Yang et al., 2024a), we directly modify their officially released code. Since our task is to reconstruct short-sequenced dynamic scenes from sparse observations (3 cameras × 4 timesteps), the original training recipes designed for long-sequenced dense views (3 cameras × 200 timesteps) are no longer appropriate. Therefore, we reduce the training iterations for all methods from 20,000 to 5,000 and linearly scale down all iteration-based hyperparameters. In our preliminary experiments, we did not observe significant differences between training for 20,000 versus 5,000 iterations, as there are only limited training views available, while training 5,000 iterations is much faster.

For generalizable approaches, LGM (Tang et al., 2024) has open-sourced their code and pre-trained models, whereas GS-LRM (Zhang et al., 2024) has not. However, LGM is originally trained on an object-centric synthetic dataset, which has a significant domain gap compared to our problem. Therefore, we followed their official code to reimplement their model within our codebase to eliminate potential misalignments due to differences in data processing, learning rate scheduling, supervision, and optimizers. For GS-LRM (Zhai et al., 2022), we implemented the model according to the descriptions provided in their paper. We train these models on the same dataset as ours with the same color, depth, perceptual and sky supervision, and the same number of iterations. Since these models do not inherently support sky processing, we modify them to predict the depth of the sky as the far plane, a predefined hyperparameter. This adjustment already enhances the performance of these methods. The number of trainable parameters of these models are controlled to be similar, *i.e.*, GS-LRM has 86.68M parameters, LGM has 103.29M parameters, while our default STORM has 100.60M parameters.

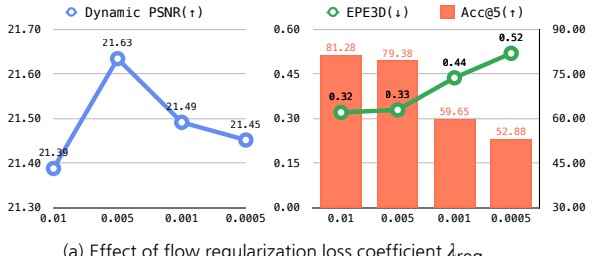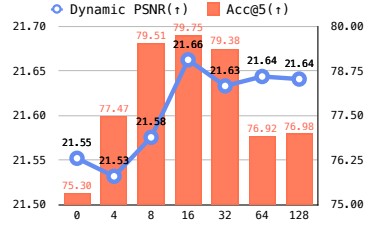

(a) Effect of flow regularization loss coefficient $\lambda_{\text{reg}}$      (b) Effect of number of motion tokens M

Figure B.1: **Ablation study on velocity regularization and motion tokens. (a)** Effect of velocity regularization coefficient $\lambda_{\text{reg}}$: We evaluate rendering quality using dynamic PSNR and flow estimation performance using EPE3D and $\text{Acc}_5$. We find that excluding this regularization frequently leads to gradient explosion and NaN loss. **(b)** Impact of motion token count: We study how the number of motion tokens $M$ influences rendering and motion estimation performance.

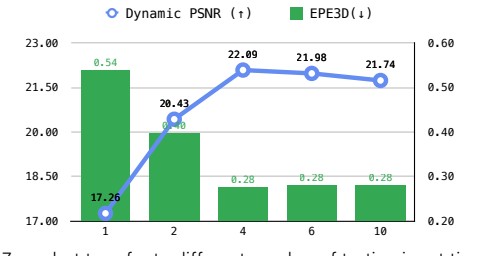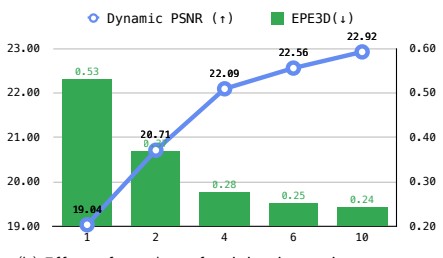

(a) Zero-shot transfer to different number of testing input timesteps      (b) Effect of number of training input timesteps

Figure B.2: **Ablation study on input timesteps. (a)** Zero-shot transfer evaluation: We test STORM pre-trained with 4 input timesteps with varying test-time input timestep configurations without re-training. **(b)** Scaling training timesteps: We train STORM with different numbers of input views to assess its adaptability to changes in input timesteps.

## B     ADDITIONAL RESULTS

### B.1    COMPARISON ON ADDITIONAL DATASETS

**NuScenes.** The NuScenes dataset (Caesar et al., 2020) contains 1000 driving scenes, each lasting 20 seconds, captured at 12Hz frame rate. These scenes are divided into 700, 150, and 150 scenes for training, validation, and testing, respectively. Similar to our evaluation protocol for the Waymo Open Dataset (Sun et al., 2020), we use 3 frontal cameras at a roughly $5.5\times$ downsampled resolution ($160 \times 288$) and leverage both `sample` (key frames) and `sweep` data. Models are trained on the training set and evaluated on the validation set with *unchanged hyperparameters*.

**Argoverse2.** The Argoverse2 dataset (Wilson et al.) contains 1,000 driving scenes, split into 700 for training, 150 for validation, and 150 for testing. It includes data from seven ring cameras, providing a 360-degree view. For training and evaluation, we use the three frontal cameras, resampled to a $192 \times 256$ resolution. The original central camera resolution is $2048 \times 1550$, while other cameras are $1550 \times 2048$, resulting in reversed aspect ratios. We do not apply special adjustments for this discrepancy and resize all images uniformly to $192 \times 256$. To simplify processing, performance is measured on full images without extracting dynamic instances. Notably, depth RMSE is higher for this dataset compared to NuScenes and Waymo, which is expected due to Argoverse2's larger sensing range of over 200m, in contrast to the approximately 80m range of the other datasets.

### B.2    ABLATION STUDY

We study the effect of different components of our method here. All experiments here are conducted on the Waymo Open Dataset. To manage the computational demands of these extensive experiments, models studied in Fig. B.1 are trained with a global batch size of 32 for 100k iterations. For the models examined in Fig. B.2-(b), we use a global batch size of 64 over the same number of iterations.

**Velocity regularization coefficient** $\lambda_{\mathrm{reg}}$**.** The impact of the velocity regularization coefficient is illustrated in Fig. B.1-(a). We observe that omitting this term often results in gradient explosion and NaN loss. Thus, this regularization term is indispensable. In this controlled setting, the optimal coefficient is found to be $5 \times 10^{-3}$, as it achieves the best dynamic PSNR and ranks second in flow estimation performance.

**Number of motion tokens** $M$**.** We evaluate the effect of the number of motion tokens on rendering and flow estimation performance in Fig. B.2-(b). When no motion token ($M = 0$) is used, the dynamic mask decoder directly predicts pixel-aligned velocities from image embeddings, keeping the number of learnable parameters nearly constant to ensure fair comparison. As shown in Fig. B.2-(b), STORM demonstrates robust performance across different motion token counts, with the best results obtained at $M = 16$.

**Number of input timesteps.** Our default configuration trains and tests STORM with 4 input timesteps. Leveraging the sequence-to-sequence nature of our Transformer-based model, we can flexibly adjust the number of input timesteps during both training and testing by appending tokens from more input timesteps or dropping tokens from existing input timesteps. This flexibility enables us to study the effect of input sparsity on STORM's performance. We conduct two ablation studies in Fig. B.2. First, we test STORM trained with 4 input timesteps under varying test-time input timestep configurations without re-training. This zero-shot transfer experiment demonstrates that STORM generalizes well to unseen input configurations, though it achieves peak performance with 4 timesteps, as expected. In the second study, we re-train STORM with different numbers of input timesteps and evaluate their performance. Results indicate that increasing the number of input timesteps improves performance. Notably, when trained with a single timestep, STORM transitions into a future prediction framework. Even in this configuration, it significantly outperforms per-scene optimization approaches that utilize 4 input timesteps for reconstruction, achieving a 2 to 4 PSNR improvement on dynamic regions (*cf.* Table 1).

## C    LIMITATIONS

While our model benefits from the scalability and flexibility of Transformer architectures, it comes with certain trade-offs. One limitation is the processed sequence length. STORM typically operates on images downsampled by a factor of 8, using inputs from three cameras and four timesteps, which results in around 7k tokens. Although we have fine-tuned STORM to handle up to 32,000 tokens in *preliminary* experiments to enable higher resolution images, longer temporal windows, or additional camera views, this comes with a non-trivial increase in computational costs for both training and inference. Another limitation is that our model requires camera intrinsic and extrinsic parameters as inputs. While these parameters are readily accessible in autonomous vehicle datasets, they may be more difficult to obtain in other domains, potentially limiting the ability to directly train and test STORM on those data domains without additional effort or preprocessing. Future works to address these limitations include better Transformer architecture with reduced complexity, joint optimization of camera parameters, and the use of geometric foundation models.

