# OpenReview forum: "STORM: Spatio-TempOral Reconstruction Model For Large-Scale Outdoor Scenes"
_ICLR.cc/2025/Conference — ICLR 2025 Poster_

### Official Review · Reviewer_HXs2 · 2024-10-31

**Soundness:** 3
**Presentation:** 3
**Contribution:** 3
**Rating:** 8
**Confidence:** 3

**Summary:**

This paper introduces a novel pipeline called STORM for dynamic scene reconstruction in real-world environments.

The core concept involves aggregating reconstructed 3D Gaussians and their motion into a unified representation, enabling the rendering of an amodal representation at any specified time step.

Additionally, to ensure STORM's effectiveness in diverse scenarios, several significant enhancements have been implemented.

Experimental results demonstrate the efficacy of the proposed method.

**Strengths:**

1. The paper is well-written and accessible, making it easy to follow the proposed approach.

2. The core idea is straightforward and well-conceived. By integrating 3D Gaussians into a unified representation, STORM enables rendering at any time step, allowing the entire network to be supervised by the color of the target time step.

3. The experiments are comprehensive and informative, demonstrating the effectiveness of the overall pipeline as well as specific enhancements, such as Latent-STORM.

**Weaknesses:**

The experiments are conducted solely on the Waymo Open Dataset, which may raise concerns about STORM’s robustness and generalizability to other datasets.

**Questions:**

The paper highlights that previous reconstruction methods rely on “dense observations across space and time,” whereas STORM “reconstructs dynamic 3D scenes from sparse, multi-timestep, posed camera images.” An insightful ablation study would involve varying the number of camera views and input time steps to assess how effectively STORM performs with different levels of view sparsity. This would provide a clearer understanding of the system's robustness under sparse observation conditions.

---

> ### Author Response · Authors · 2024-11-22
> **Response to Reviewer HXs2**
>
> Thank you for your thoughtful review and constructive feedback. We appreciate your recognition of the strengths of our work, and we address your concerns and questions in detail below. We hope these clarifications demonstrate the robustness and versatility of STORM.
>
> **Q1: Generalizability Beyond the Waymo Open Dataset**
>
> **A:** We have addressed this point in detail in the **general response**. In summary, we evaluated STORM on two additional datasets—NuScenes and Argoverse2—to validate its generalizability. These results, included in the revised manuscript, demonstrate STORM’s robustness and versatility across diverse real-world scenarios. For further details and results, please refer to the **general response**.
>
> **Q2: Robustness to Sparse Observations**
>
> **A:**
> This is a very insightful and inspiring suggestion that motivates us to explore more. We believe showing this study will further highlight the flexibility of our approach. Now in the revised manuscript, we investigated STORM's robustness to varying numbers of input timesteps. Leveraging the sequence-to-sequence nature of our Transformer-based model, STORM can flexibly adapt to different numbers of input timesteps during both training and testing. We conducted two ablation studies (detailed in Fig. B.2):
> - (1) Testing STORM trained with 4 input timesteps on different number of test-time input timesteps (e.g. 1, 2, 6, 10) without re-training (zero-shot transfer). Results show that STORM generalizes well to unseen configurations, with peak performance achieved at 4 timesteps, as expected.
> - (2) Retraining STORM with different number of input timesteps (including 1, 2, 4, 6, 10). Results indicate that increasing timesteps improves performance. Notably, when trained with *a single timestep*, STORM generalizes to a future prediction framework (by being able to predict Gaussian velocities given a single timestep), achieving a remarkable **2–4 PSNR** improvement on dynamic regions compared to per-scene optimization methods using *4 timesteps*.
> These ablations highlight STORM’s flexibility and robustness under varying conditions, further validating its applicability to real-world scenarios.
>
> These ablation studies highlight STORM’s robustness to sparse observations and varying training time and test-time input timesteps. For additional details, please refer to the revised manuscript.
>
> We hope these updates and clarifications address your concerns and further demonstrate the robustness, versatility, and real-world applicability of STORM.
>
> Thank you again for your valuable feedback, which has significantly improved the manuscript!
>
> Please let us know if there are additional points we can address.

---

> ### Author Response · Authors · 2024-11-25
> **Follow-Up on Rebuttal Feedback**
>
> Dear Reviewer HXs2,
>
> Thank you for your thoughtful and constructive review of our submission. Your feedback has been instrumental in strengthening our work, particularly regarding generalizability and robustness under sparse observations.
>
> With the discussion deadline approaching, we want to ensure that all your concerns are fully addressed. Please do not hesitate to reach out if you require further clarifications or analyses.
>
> To address your concerns:
> - We conducted experiments on NuScenes and Argoverse2, demonstrating STORM’s applicability across datasets.
> - We included new ablation studies that explore performance under varying numbers of input timesteps, highlighting robustness to sparse observations and different training/testing configurations. We highly appreciate your insightful review of this point!
>
> We believe these updates address your concerns and would greatly appreciate it if you could review the revisions. Your confirmation or additional feedback would be highly valued.
>
> Additionally, we hope you could consider raising your score based on our rebuttal, as we believe these revisions, guided by your thorough review, have greatly improved the paper.
>
> Thank you again for your time and valuable suggestions!
>
> Best regards,
>
> Authors

---

> > ### Comment · Reviewer_HXs2 · 2024-11-26
> >
> > The authors' response has effectively addressed my concerns. I am increasing my rating to 8.

---

> > > ### Author Response · Authors · 2024-11-27
> > > **Thank you!**
> > >
> > > Dear Reviewer HXs2,
> > >
> > > We are truly grateful for your constructive feedback in the first round and for increasing your rating after seeing our rebuttal.
> > >
> > > Your insights have been instrumental in improving our work, and we deeply appreciate the time and effort you dedicated to reviewing our submission!
> > >
> > > Thank you!
> > >
> > > Best regards,
> > >
> > > The Authors

---

### Official Review · Reviewer_N7nA · 2024-11-03

**Soundness:** 3
**Presentation:** 3
**Contribution:** 3
**Rating:** 6
**Confidence:** 4

**Summary:**

The paper presents STORM, a spatio-temporal reconstruction model that uses a data-driven Transformer architecture to accurately reconstruct dynamic outdoor scenes from sparse camera images without explicit motion supervision. It employs 3D Gaussian Splatting to aggregate per-frame data into an amodal, cohesive representation, enabling consistent scene reconstruction over time. The model is self-supervised, relying on reconstruction loss and incorporating motion tokens to capture motion primitives efficiently. Extensive testing on the Waymo Open Dataset shows that STORM outperforms traditional methods in photorealism and speed, achieving real-time performance within 0.2 seconds per clip.

**Strengths:**

1. The paper presents a clear and well-organized approach to addressing the defined problem of dynamic scene reconstruction.
2. The proposed method achieves state-of-the-art performance, showcasing its practical effectiveness and potential.
3. The video results available on the linked website are highly illustrative, providing clear insights into both the strengths and limitations of the proposed method.

**Weaknesses:**

1. The evaluations are limited to the Waymo dataset, which raises concerns about the comprehensive validation of the method. To strengthen the claims, it would be beneficial to demonstrate the effectiveness of the model across multiple datasets.
2. While the paper targets large-scale outdoor scene reconstruction, existing works such as [1] also address similar challenges. If the primary focus is on dynamic scene reconstruction, the paper’s title should explicitly include the term ‘dynamic,’ and comparative results with dynamic 3D reconstruction methods like [2] should be included to provide context and highlight advancements.
3. One of the main contributions is the introduction of 'motion tokens' in the Transformer’s input sequence, alongside the unique loss function components (reconstruction loss, sky loss, and velocity regularization loss). However, the absence of ablation studies makes it difficult to discern the effectiveness of these specific modules. Adding ablation studies would enhance the paper by showing the impact of each proposed component.

[1] Lin, Jiaqi, et al. "Vastgaussian: Vast 3d gaussians for large scene reconstruction." *Proceedings of the IEEE/CVF Conference on Computer Vision and Pattern Recognition*. 2024.
[2] Fischer, Tobias, et al. "Dynamic 3D Gaussian Fields for Urban Areas." *arXiv preprint arXiv:2406.03175* (2024).

**Questions:**

1. Could the authors include comparisons across multiple datasets to validate the generalizability of the proposed method?
2. Could the authors provide ablation studies to demonstrate the effectiveness of each proposed module in the method?

---

> ### Author Response · Authors · 2024-11-22
> **Response to Reviewer N7nA**
>
> We thank the reviewer for the detailed review and thoughtful feedback. We have incorporated the suggestions into the revised manuscript and addressed each of the raised questions and concerns below. We hope these clarifications effectively demonstrate the robustness and contributions of our work.
>
> **Q1: Generalizability Across Multiple Datasets**
>
> **A:**
> We have addressed this point in detail in the **general response**. In summary, we evaluated STORM on two additional datasets—NuScenes and Argoverse2—to validate its generalizability. These results, included in the revised manuscript, demonstrate STORM’s robustness and versatility across diverse real-world scenarios. Please refer to the **general response** for further details and results.
>
> **Q2: Dynamic Scene Reconstruction Focus**
>
> **A:**
> We acknowledge the relevance of prior works, such as VastGaussian [1] and Dynamic 3D Gaussian Fields [2], and have added references to these methods in the revised manuscript. Below, we explain their relationship to our work:
> - VastGaussian [1]: This work focuses on improving the memory efficiency and scalability of per-scene optimized 3D Gaussian Splatting. While this is orthogonal to our goal of real-time, feed-forward dynamic reconstruction, we believe STORM can benefit from the techniques proposed in VastGaussian in future research.
> - Dynamic 3D Gaussian Fields [2]: This method leverages bounding box annotations to learn dynamic objects. Since STORM does not use label information, we have focused on comparing state-of-the-art methods that similarly do not require annotations. For this reason, we have not included this approach in our current baselines.
>
> **Title Clarification**：Regarding the title, while we initially emphasized the temporal aspect of reconstruction, we agree that explicitly including “dynamic” in the title would better highlight STORM’s focus and will consider this revision.
>
> **Q3: Ablation Studies on Proposed Components**
>
> Thank you for highlighting the need for ablation studies. We have conducted detailed ablation analyses on key components, including velocity regularization, motion token count, and input timesteps. For a full summary of these results, please refer to the **general
> response**. These studies further validate STORM’s robustness and the effectiveness of its design choices.
>
> We hope these updates and clarifications address your concerns and demonstrate STORM’s strengths, versatility, and real-world applicability. Thank you again for your valuable feedback, which has greatly improved our work. Please let us know if there are any additional points we can address.
>
>
> [1] Lin, Jiaqi, et al. "VastGaussian: Vast 3D Gaussians for Large Scene Reconstruction." CVPR 2024.
> [2] Fischer, Tobias, et al. "Dynamic 3D Gaussian Fields for Urban Areas." arXiv preprint arXiv:2406.03175 (2024).

---

> ### Author Response · Authors · 2024-11-25
> **Follow-Up on Rebuttal Feedback**
>
> Dear Reviewer N7nA,
>
> Thank you for your detailed and thoughtful review of our submission. Your feedback on generalizability, comparisons, and ablation studies has been immensely helpful in refining our work.
>
> With the discussion deadline approaching, we want to ensure that all your concerns are fully addressed. Please do not hesitate to reach out if you require further clarifications or analyses.
>
> In our rebuttal and revised manuscript, we have addressed your points:
> - Generalizability has been demonstrated with additional experiments on NuScenes and Argoverse2, where STORM achieved state-of-the-art performance again, validating STORM’s robustness across diverse datasets.
> - References to related works, such as VastGaussian and Dynamic 3D Gaussian Fields, have been added for context.
> - Comprehensive ablation studies now evaluate the impact of motion tokens, velocity regularization, and other components.
>
> We hope these updates and clarifications have addressed your concerns and strengthened our submission. If you have had a chance to review our revisions, we would greatly appreciate your feedback or confirmation on whether the updated manuscript meets your expectations. Your confirmation or additional comments would mean a great deal to us.
>
> Additionally, we hope you could consider raising your score based on our rebuttal, as we believe these revisions, guided by your thorough review, have greatly improved the paper.
>
> Thank you once again for your time, thoughtful feedback, and support.
>
> Best regards,
>
> Authors

---

> > ### Comment · Reviewer_N7nA · 2024-11-25
> >
> > Thank you for the response and addressing my concerns.
> > I increased my initial rating of 5 to 6.

---

> > > ### Author Response · Authors · 2024-11-27
> > > **Thank you!**
> > >
> > > Dear Reviewer N7nA,
> > >
> > > Thank you for your constructive feedback in the first round and for raising your rating after reviewing our rebuttal.
> > >
> > > Please feel free to reach out if there are any further improvements or concerns we can address --- we are committed to delivering the strongest possible work. Your feedback matters a lot to us.
> > >
> > > Thank you!
> > >
> > > Best regards,
> > >
> > > The Authors

---

### Official Review · Reviewer_JKrr · 2024-11-03

**Soundness:** 3
**Presentation:** 3
**Contribution:** 3
**Rating:** 6
**Confidence:** 3

**Summary:**

This paper introduces STORM, a spatio-temporal reconstruction model for rendering dynamic outdoor scenes with sparse images. Combining image tokens, ray tokens, and time tokens, STORM uses a Transformer to predict attributes of 3D Gaussians (position, rotation, scale, opacity, color, velocity) at each timestep for real-time rendering. The training requires ground truth images, depth map and predicted sky mask from pretrained segmentation model.

**Strengths:**

The paper is clearly written.

STORM may be the first method that combines feed-forward 3D Gaussian Splatting and dynamic scenes.

STORM achieves better rendering quality and scene flow accuracy than baselines.

**Weaknesses:**

STORM only evaluates the quantitative performance in short video clips (about 2 seconds on Waymo open dataset). Applying STORM on 20-second videos requires processing in multiple video clips (L428) because of the heavy computation, which is also discussed in limitations.

There is no quantitative ablation study.

**Questions:**

In Fig.3, can authors add ground truth images for comparison? Based on current visualization, Latent-STORM sometimes performs better and sometimes performs worse than STORM.

Could authors provide some reference for handling exposure mismatches with affine transformation?

---

> ### Author Response · Authors · 2024-11-22
> **Response to Reviewer JKrr**
>
> We thank the reviewer for the thoughtful review and constructive comments. We address the questions and concerns below and have incorporated additional clarifications and updates into the revised manuscript. We hope that our revisions and this rebuttal address the reviewer's concerns.
>
> **Q1. Short video sequence v.s. Long video sequence.**
>
> **A:**
> - **Computation efficiency**. We would like to emphasize the computational efficiency of STORM. Applying STORM to a 20-second video sequence requires approximately **2 seconds** of processing time, compared to **hours** of training time required by per-scene optimization methods. This represents a significant improvement in efficiency. STORM is already capable of handling 64 or more input views during training with our current implementation.
>     - To further address your concerns, we trained a STORM variant with 10 timesteps × 3 views (30 input views in total) spanning a 10-second duration (views evenly spaced at 1-second intervals). This configuration achieves scene coverage up to 100m. A video demo illustrating this capability is available here: https://anonymousi079j.github.io/STORM_review/assets/videos/10s_demo.mp4. Note that we did not change any hyperparameters for this variant so this example is purely for illustration.
>     - Additionally, STORM is expected to benefit from advanced techniques such as FlashAttentionV3, building on our current implementation, which utilizes FlashAttentionV2. These advancements could further extend our model's context length and scalability.
> - **Autonomous vehicles (AV) application**. For AV scenarios, long rollouts (e.g., 20 seconds) are rarely necessary for training or testing. Typically, adversarial scenarios in AV testing require shorter sequences (e.g., 4-5 seconds). STORM’s design aligns well with these practical requirements, making it a compelling fit for AV applications.
>
>
> **Q2. Quantitative ablation.**
> Thank you for emphasizing the importance of quantitative ablations. We have conducted comprehensive ablation studies analyzing the contributions of key components, including velocity regularization, motion token count, and input timesteps. These findings are summarized in the general response and detailed in the revised manuscript (Line 429-460 and Appendix B.2 (Figures B.1–B.2)). Please refer to the general response for a full overview of these results, which highlights STORM’s robustness and flexibility under varying configurations.
>
> **Q3. Side-by-side visual comparison.**
> We appreciate this suggestion to include side-by-side comparisons. To address this, we have updated Figure 3 to include a pointer to our project page, where readers can access side-by-side video comparisons (see the “Human Modeling with Latent-STORM” section). Due to space constraints, ground truth images were not included in the manuscript but can be added in the camera-ready version if space permits.
>
> **Q4: Reference to Handling Exposure Mismatches with Affine Transformation**
> The revised manuscript now references the "Urban Radiance Fields" [1] work, which employs learnable affine transformations to align exposure differences across cameras in a per-scene optimization setting. This provides additional context for our approach.
>
>
> We hope these updates and clarifications address your concerns and demonstrate STORM’s strengths, versatility, and real-world applicability. Thank you again for your valuable feedback, which has greatly improved our work. Please let us know if there are any additional points we can address.
>
>
> [1] Rematas, Konstantinos, et al. "Urban radiance fields." Proceedings of the IEEE/CVF Conference on Computer Vision and Pattern Recognition. 2022.

---

> ### Author Response · Authors · 2024-11-25
> **Follow-Up on Rebuttal Feedback**
>
> Dear Reviewer JKrr,
>
> Thank you for your constructive feedback on our submission. Your comments on long sequences, ablation studies, and visualization have been instrumental in refining the manuscript.
>
> With the discussion deadline approaching, we want to ensure that all your concerns are fully addressed. Please do not hesitate to reach out if you require further clarifications or analyses.
>
> To address your points:
> - We expanded our experiments to demonstrate STORM's scalability to more input frames or longer sequences, with new video demos included on the project page.
> - Detailed ablation studies have been added to validate the contributions of individual components.
> - Visualizations now include a link to a side-by-side video comparison for STORM v.s. Latent-STORM on human modeling.
>
> We hope these updates address your concerns. If you have had a chance to review our revisions, we would greatly appreciate your feedback or confirmation on whether the updated manuscript meets your expectations.
>
> Additionally, we hope you could consider raising your score based on our rebuttal, as we believe these revisions, guided by your thorough review, have greatly improved the paper.
>
>
> Thank you once again for your time and insights.
>
> Best regards,
>
> Authors

---

> > ### Comment · Reviewer_JKrr · 2024-11-27
> >
> > Thanks for the response and additional experiments on other datasets. I will keep my positive rating.

---

### Official Review · Reviewer_Wu7P · 2024-11-03

**Soundness:** 3
**Presentation:** 3
**Contribution:** 3
**Rating:** 6
**Confidence:** 4

**Summary:**

This paper proposes a method for dynamic 3D reconstruction (4D reconstruction) of urban scenes from temporally and spatially sparse video input.

The model is based on a per-frame transformer architecture that takes image, ray (from given camera intrinsics and extrinsics), time and motion latent tokens as input as well as some special tokens for handling sky and varying camera exposure. The model takes tokens from an image coming from single camera v and timestamp t, and then processes these tokens jointly with a self-attention transformer, to output feature maps F^v_t for that image. These tokens are then decoded into pixel-aligned 3D Gaussian Splatting parameters G^v_t, which are used to reconstruct the input frame as well as to compute the depth map. In addition, the motion tokens are decoded into a set of motion queries and velocity bases, which are employed to estimate the velocity vector of each 3D Gaussian.

Given a set of training frames coming from one or several cameras, the model is run on each frame independently to obtain independent sets of Gaussians. These independent Gaussians can then be warped to any particular timestamp using the velocity estimates, and assuming velocity constancy.

In addition to the vanilla model, the authors propose a latent variant where instead of predicting the Gaussian parameters directly, the transformer first predicts higher dimensional vectors on a lower-dimensional grid, which are then upsampled to full-resolution using a CNN decoder.

The model is trained and evaluated on clips from the Waymo Open Dataset, using the provided camera intrinsics and extrinsics, and three frontal car cameras.

The authors evaluate their method against both per-scene optimization and feed-forward methods. For each 20 frame test clip, 4 frames are provided as inputs for reconstruction to each baseline, and the rest are used for evaluation. The results in Table 1 show that the proposed method obtains SOTA performance on reconstruction metrics compared to both per-scene optimization and feed-forward methods. In addition, the authors evaluate their method on the scene flow estimation task, also obtaining SOTA results.

Finally, the authors link a website with numerous quantitative results which allow to better assess the performance of the proposed method.

**Strengths:**

This is an interesting method which applies per-frame feed-forward 3DGS models to a multi-camera scenario for the urban driving setting. While similar methods exist for object-centric scenes (Splatter Image, LatentSplat) and static objects, or even object-centric scenes and dynamic objects (4DGS Guanjun Wu et al.), I believe the extension to outdoor driving scenes is novel and relevant. The authors successfully devised strategies to deal with the complexities of such scenes, by explicitly modeling the sky and the differences in camera exposure.

Furthermore, the authors adopt a similar strategy for modeling dynamics through scene flow as in NSFF, but which results to be more tractable due to the usage of 3DGS as a discrete representation of the scene. This methods seems to be effective in modelling short term dynamics.

Quantitative and qualitative results show that the method achieves good new-view synthesis results from only a few seen views.

The proposed latent-STORM variant shows improved reconstruction of local details, such as for the articulated bodies of walking pedestrians.

**Weaknesses:**

Some details of the method are not very clear from the paper. For instance, it's not clear how the aggregation and transformation processes are applied on the predictions of the 4 training frames for generating the predicted images \hat{I}.

In addition, it's not clear whether eq. (4) is correct and how exactly the weighs w are computed as the exact shapes of q and k are not specified. Also, it's not clear how the two different estimates v_t^{-} and v_t^{+} from eq (1) are obtained from what is described in eqs. (3,4). Furthermore, not many details are given about how these weights can be used to generate the motion segmentation figures, and the impact of the number of motion basis. It's also not very clear how the background scene is treated during this motion estimation.

Regarding the quantitative results, it's not clear if the baselines for Table 2 have been retrained specifically on the same training set for fairness. Also, it's specified that the baseline models NSFP and NSFP++ require LiDAR input, but it's not clear whether these methods also benefit from having 3 different input views.

Regarding the qualitative results, having 3D visualizations in addition to depth-maps would help further assessing the temporal consistency and geometric correctness of the predictions.

Finally some of the proposed applications or extensions in sec. 4.3 are not explained in sufficient detail. For instance, how the point trajectories are chained for point tracking, given that the local scene-flow estimations are sparse and don't cover the whole space. In addition, it's not clear how latents can be manipulated in the latent-STORM model for achieving the edition, control and inpainting effects.

**Questions:**

- Please clarify how the images \hat{I} are computed for a given training batch and which aggregations and transformations are involved.

- Please clarify the shapes of q and k in eq. (4) and how the weights are used for generating the motion segmentation visualizations, as well as how the number of motion basis is determined.

- Please clarify how the baselines are trained and which inputs they require.

- If possible, having 3D visualizations of the results would be beneficial.

- Please provide additional details about how point-tracking can be obtained, and how the editing, control and inpainting behaviours can be obtained in latent-STORM.

---

> ### Comment · Reviewer_Wu7P · 2024-11-21
> **No clarifications**
>
> Unfortunately the authors have not produced any clarifications of the points addressed by reviewers. In this situation, I'm inclined to reject this paper which, while interesting, has a number of issues that remain unanswered.

---

> > ### Author Response · Authors · 2024-11-22
> > **Rebuttals posted**
> >
> > Dear Reviewer Wu7P,
> >
> > Thank you for your thoughtful comments and for taking the time to review our work. We truly appreciate your feedback and the opportunity to address your concerns.
> >
> > We sincerely apologize for the delay in providing clarifications. To ensure a comprehensive response, we focused on completing additional experiments and making revisions to the manuscript. We've now submitted our rebuttal and updated the paper with additional experiments and improved clarity to address the points raised. We kindly invite you to review these updates and share any further thoughts you might have.
> >
> > We understand your concerns and are committed to ensuring clarity and rigor in our work. Please let us know if our rebuttal resolves your concern. Thank you again for your time and valuable feedback!
> >
> > Best regards,
> > The Authors

---

> ### Author Response · Authors · 2024-11-22
> **Response to Reviewer Wu7P (1/2)**
>
> We sincerely thank the reviewer for the insightful comments. We have incorporated these suggestions into our revised manuscript and provided detailed explanations for each of the reviewer's questions below. We hope that our revisions and this rebuttal address the reviewer's concerns.
>
> In addition, *we will open-source our code* for the community to examine the implementation details more closely.
>
> **Q1: How are the images $\hat{I}$ computed for a given training batch based on aggregation and transformation?**
>
> **A:** Thank you for highlighting the need for clarity. We apologize for the lack of clarity in our original manuscript and appreciate the opportunity to clarify these. Here is a step-by-step explanation:
>
> 1. **Input Batch Structure**: We describe the model’s behavior from a batch perspective.
> - **Context timesteps ($t$)**: These are the input timesteps from which we predict per-frame 3D Gaussians (3DGS) and velocities. For four context frames, $t$ has the shape $(b, \text{ctx}_t)$, where $b$ is the batch size and $\text{ctx}_t = 4$.
> - **Target timesteps ($t'$)**: These are the timesteps at which we render $\hat{I}$, with shape $(b, \text{tgt}_t)$.
>
> 2. **Computing Time Differences**: We calculate *dense* time differences between target and context timesteps: $\text{tdiff} = t' - t$, resulting in a tensor of shape $(b, \\text{tgt}_t, \\text{ctx}_t)$. The pseudo-code is `tdiff = tgt_time.unsqueeze(-1) - ctx_time.unsqueeze(-2)`. This corresponds to the $( t' - t)$ term in Equation (1) of our paper.
>
> 3. **Preparing Velocity and Center Tensors**:
> - **Velocities ($\mathbf{v}_t$)**: Predicted for each context frame, shape $(b, \text{ctx}_t, N, 6)$, where $N= v\times H \times W$ is the number of 3D GS at each timestep, and $v, H, W$ denotes the number of camera views, and the height and width of input images.
>
> - **Centers ($\boldsymbol{\mu}_t$)**: Gaussians' centers, shape $(b, \text{ctx}_t, N, 3)$.
>
> - We expand these tensors to align with target timesteps: $\mathbf{v}_t \rightarrow (b, \text{tgt}_t, \text{ctx}_t, N, 6)$ and $\boldsymbol{\mu}_t \rightarrow (b, \text{tgt}_t, \text{ctx}_t, N, 3)$
>
> 4. **Applying the Transformation**:
> - With all tensors now sharing the shape $(b, \text{tgt}_t, \text{ctx}_t, \ldots)$, we compute Equation (1) using element-wise operations.
> - After applying Equation (1), we obtain the transformed centers $\boldsymbol{\mu}_{t' \rightarrow t}$ with shape $(b, \text{tgt}_t, \text{ctx}_t, N, 3)$. We simplify this notation to $\boldsymbol{\mu}'_t$ because `OpenReview` suffers from formula display issues when using excessive subscripts.
>
> 5. We aggregate context to form the amodal representation: $\mathcal{G}_{t'} = \bigcup\_t  \boldsymbol{\mu}'_t$, resulting in a shape of $(b, \text{tgt}_t, \text{ctx}_t \times N, 3)$. Here, the third dimension $\text{ctx}_t \times N$ corresponds to the "amodal" representation we define in the manuscript, which combines all Gaussians from different context frames into a single set for each target timestep.
>
> 6. We render $\hat{I}_{t'}$ at each target timestep $t'$ by projecting the aggregated Gaussians at this timestep (i.e., selecting one sample from the second dimension of  $(b, \text{tgt}_t, \text{ctx}_t \times N, 3)$) onto the image plane of the desired viewpoint.
>
> -----
>
> **Q2: What are the shapes of $q$ and $k$ in Equation (4), and how are the weights used for generating the motion segmentation visualizations?**
>
> **A**:
> - **Shapes of $q, k$**: We apologize for the ambiguity in the original manuscript. The updated paper now includes additional clarifications (Lines 232–253). In brief:
>     - $q \in \mathbb{R}^{M \times e'}$ and $k \in \mathbb{R}^{H \times W \times e'}$, where $M$ is the number of motion tokens, $H$ and $W$ are the height and width of the input images, $e’=32$ is a low-dimensional motion embedding space (mentioned in Line 916) used to measure the dot-product similarity between queries and keys.
>     - For further details, we strongly encourage the reviewer to refer to the updated manuscript, which provides a clearer and more comprehensive explanation.
>
> - **Motion Segmentation Visualization**:
> In the updated Equation (4), we clarify that the motion assignment weights $\mathbf{w} \in \mathbb{R}_{+}^{H \times W \times M}$ are in the range $(0, 1)$.
> In the updated Lines 504–506, we clarify that the motion segmentation mask is derived by applying an `argmax` operation on the per-pixel assignment weights $\mathbf{w}$ along the motion token dimension $M$.
>
> - **Scene Background**: Thank you for raising this insightful point. To clarify, we do not explicitly differentiate between background and foreground. Static regions may be over-segmented into separate groups since the assignment process is purely self-supervised. Investigating how these static groups form—whether by color, geometry, or semantics—would be an intriguing direction for future work.

---

> > ### Author Response · Authors · 2024-11-22
> > **Response to Reviewer Wu7P (2/2)**
> >
> > **Q3. Clarification on Baselines**
> >
> > **A:**
> > - **Rendering**: We appreciate the request for further clarification. Implementation details for all baselines were already provided in the Appendix of *initial* manuscript (now it has been relocated to Appendix A.3 of the revised manuscript). To enhance clarity, we have expanded these descriptions. In short, **all these baselines are trained on our datasets** with the same number of iterations and similar supervision signals (RGB, depth, and sky). If the reviewer has specific questions or would like additional details, we are happy to provide further information.
> >
> > - **Flow estimation**: We seek to address any ambiguity regarding this comparison. For NSFP and NSFP++, we provide LiDAR points corresponding to the 3 frontal cameras (if this refers to the “3 different input views”). However, if the concern is about input timesteps (e.g., STORM uses 4 timesteps, while NSFP and NSFP++ use 2), we note that this difference is inherent to the design of these compared methods. Extending NSFP/NSFP++ to accept more than two timesteps is non-trivial and remains a research question, as these methods are fundamentally limited to their original pairwise input configuration.
> >
> >
> > -----
> >
> > **Q4. 3D visualization**
> >
> > **A:** We have updated our project page to include 4D (3D+time) visualizations to assess temporal consistency and geometric correctness. Please refer to the "4D Visualization" Section at https://anonymousi079j.github.io/STORM_review/.
> > These additions include point-tracking visualizations across time. These demos are presented as video recordings due to time constraints. We plan to make them interactive in the future.
> >
> > -----
> > **Q5. Detailed explanation for different applications.**
> >
> > **A:**
> > - **Point tracking**: Thank you for raising this question. Point tracking is achieved by chaining local scene-flow estimations across consecutive frames. Specifically, at each frame t, we use the predicted scene flow to transform the Gaussians to their estimated positions in the next frame t+1. For every Gaussian in frame t+1, we identify its nearest transformed Gaussian from frame t and connect them to form the trajectory. This process is recursively applied across all frames to construct the complete trajectories. We will also release our visualization code for the community to check more details.
> > - **Sparsity of local scene-flow estimations**: There seems to be a misunderstanding regarding this point. STORM estimates dense **per-pixel** scene flow that encompasses *all visible points* in the space observed by the context frames. However, it does not cover regions that are invisible in all frames. The trajectories may appear sparse in our manuscript because we uniformly subsample them to reduce visualization overhead. We are happy to add further clarifications on this aspect in the revised manuscript if needed.
> > - **Editing latent-STORM**: We have added an explanation in Lines 511–514 of the revised manuscript. The image generation process in Latent-STORM involves:
> >     1. **Predicting Patch-Aligned Latent Gaussians**: A single latent Gaussian will be predicted from each patch of the image. These Gaussians form a "set".
> >     2. **Projecting onto 2D Latent Feature Maps**: The latent Gaussians from this set are projected onto the 2D plane to form latent feature maps.
> >     3. **Decoding**: The feature maps are passed through a decoder to reconstruct color images and depth maps.
> >
> > Since the latent Gaussians represent physical particles in space—with attributes like coordinates, scale, opacity, and rotation—we can manipulate them. For instance, by removing certain latent Gaussians *from the set* before projection, the corresponding objects are removed from the 2D feature map, and the decoder will not reconstruct them. This enables editing, control, and inpainting capabilities!
> >
> > ----
> > We hope these detailed explanations address your concerns. Please let us know if you have additional questions or require further clarification. Your suggestions and feedback are greatly appreciated, and we are always happy to address any concerns.

---

> ### Author Response · Authors · 2024-11-25
> **Follow-Up on Rebuttal Feedback**
>
> Dear Reviewer Wu7P,
>
> Thank you for your thoughtful and detailed feedback on our submission. Your insights have been invaluable in refining our work, and we have addressed your comments comprehensively in our rebuttal.
>
> With the discussion deadline approaching, we want to ensure that all your concerns are fully addressed. Please do not hesitate to reach out if you require further clarifications or analyses.
>
> We understand you had concerns regarding some ambiguities in the initial submission. To address these, we have:
> - Clarified how images $\hat{I}$ are computed and improved the design explanations of the *motion keys and queries*, with details provided in the updated manuscript.
> - Conducted comprehensive ablation studies on key components, including *the effect of the number of motion bases*.
> - Provided additional details to complement our initial description of baseline implementation details.
> - Made 3D visualizations available on our project page to better illustrate temporal consistency and geometric accuracy.
> - Clarified how point-tracking is achieved and explained the editing process in latent-STORM.
>
> We hope these updates and clarifications have addressed your concerns and strengthened our submission.
>
> If there are any further clarifications or analyses you would like us to provide, please let us know.
>
> Additionally, we hope you could consider raising your score based on our rebuttal, as we believe these revisions, guided by your thorough review, have greatly improved the paper.
>
> Thank you again for your thoughtful feedback and for helping us improve this work!
>
> Best regards,
>
> Authors

---

> ### Author Response · Authors · 2024-12-02
> **Follow-Up on Review Feedback**
>
> Dear Reviewer Wu7P03,
>
> Thank you once again for your thoughtful feedback on our paper and the time you’ve dedicated to reviewing it.
>
> We noticed that there is only one day left in the discussion period, and we would greatly appreciate any additional feedback you might have. If there are clarifications, further analyses, or experiments you’d find valuable, please let us know. We are eager to address any remaining concerns or suggestions to strengthen our work further.
>
> We truly value the effort and insight you’ve already contributed and look forward to your follow-up response, including any reflection on the current score.
>
> Thank you again for your time and support in this process. Please don’t hesitate to reach out with any further questions or thoughts.
>
> Best regards,
>
> The Authors

---

### Author Response · Authors · 2024-11-22
**General response**

We sincerely thank all reviewers for their detailed and constructive feedback, which has been invaluable in refining and improving our work. We are encouraged by the recognition that STORM represents the **first** feed-forward 4D scene reconstruction method for dynamic outdoor environments, a novel and promising direction. Reviewers commended STORM’s state-of-the-art performance in rendering quality and scene flow accuracy, the novel and grounded integration of 3D Gaussian Splatting, and the clear presentation of its pipeline. Below, we summarize the key updates made in response to reviewers' feedback. Our manuscript has been updated accordingly with all major revisions **marked in blue**. Meanwhile, we put the link to our anonymous project page here for easy accessibility: https://anonymousi079j.github.io/STORM_review/

### **1. Generalizability Across Datasets**

Reviewers `N7nA` and `HXs2` highlighted the importance of validating STORM’s robustness beyond the Waymo Open Dataset. To address this, we have extended our experiments to include evaluations on two additional datasets: NuScenes [1] and Argoverse2 [2].
STORM achieves state-of-the-art performance compared to other feedforward reconstruction methods on both datasets in terms of full-image PSNR and Depth RMSE, as shown in Table 2 of the revised manuscript. We summarize the results as below for convenience:
| Method   | NuScenes PSNR ↑ | NuScenes D-RMSE ↓ | Argoverse2 PSNR ↑ | Argoverse2 D-RMSE ↓ |
|----------|------------------|-------------------|--------------------|---------------------|
| LGM      | 23.21           | 7.34              | 22.93             | 14.20              |
| GS-LRM   | 24.53           | 7.71              | 24.49             | 14.70              |
| Ours     | **24.90**       | **5.43**          | **24.80**         | **13.51**          |


These results validate STORM’s robustness and versatility across diverse real-world scenarios.

We have also added 4D visualizations (3D points and point trajectories) for the Waymo, NuScenes, and Argoverse2 datasets to the project page. Please refer to the “4D Visualization” section in our project page for details.

### **2. Ablation**
We appreciate the reviewers’ requests for a deeper analysis of STORM’s components and its robustness under sparse observation conditions. In response, we conducted comprehensive ablation studies, summarized below and detailed in Appendix B.2 (Figures B.1–B.2):
- **Velocity regularization coefficient $\lambda_\text{reg}$**: Omitting this term causes gradient instability and NaN loss, making it indispensable. In our ablation, the optimal coefficient value, $5 \times 10^{-3}$, achieves the best balance between dynamic PSNR and flow estimation performance.
- **Motion token count ($M$)**: We evaluated performance with varying numbers of motion tokens ($M=0, 4, 8, 16, 32, 64, 128$. Results show that STORM performs robustly across a range of motion token counts, with the best results achieved at $M=16$. We have updated all our quantitative results in our tables to these best hyperparameter settings ($\lambda_\text{reg}=5 \times 10^{-3}, M=16$) in the revision.
- **Number of Input timesteps:** Encouraged by **Reviewer `HXs2`**, we investigated STORM's robustness to varying numbers of input timesteps. Thanks to the sequence-to-sequence nature of our Transformer-based model, STORM allows flexibility in choosing the number of input timesteps during both training and testing. We conducted two ablation studies (detailed in Fig. B.2):
    - (1) Testing STORM trained with 4 input timesteps on different number of test-time input timesteps (e.g. 1, 2, 6, 10) without re-training (zero-shot transfer). Results show that STORM generalizes well to unseen configurations, with peak performance achieved at 4 timesteps, as expected.
    - (2) Retraining STORM with different number of input timesteps (including 1, 2, 4, 6, 10). Results indicate that increasing timesteps improves performance. Notably, when trained with *a single timestep*, STORM generalizes to a future prediction framework (by being able to predict Gaussian velocities given a single timestep), achieving a remarkable **2–4 PSNR** improvement on dynamic regions compared to per-scene optimization methods using *4 timesteps*.
These ablations highlight STORM’s flexibility and robustness under varying conditions, further validating its applicability to real-world scenarios.

### **3. Clarifications**
- **Technical Clarifications**: We revised the manuscript to clarify key technical details, including improved notations for motion group assignments, a brief illustration of how to obtain point trajectories and explanations of latent-STORM editing.
- **References and Related Work**: We have added references to all related works mentioned by reviewers.
- **Visualization Updates**: Our anonymous project page has been updated with additional 3D and tracking visualizations, providing further insights into STORM’s performance.

---

### Meta-Review · Area_Chair_mxTp · 2024-12-21

**Metareview:**

This paper introduces a novel pipeline for dynamic scene reconstruction, which addresses several challenges inherent in modeling dynamic scenes with high fidelity and accuracy. By leveraging a novel integration of 3D Gaussians into a unified representation, the method achieves efficient rendering and reconstruction, enabling supervision directly through the reconstruction loss.

Reviewer Wu7P initially expressed concerns about the clarity of certain implementation details and experimental settings. These issues were adequately clarified in the rebuttal, where the authors provided more detailed descriptions of their methodology. Reviewer JKrr noted the lack of a quantitative ablation study in the initial submission, which was a valid critique. In response, the authors included comprehensive ablation studies in their rebuttal, highlighting the contributions of each component in their pipeline. Both Reviewers N7nA and HXs2 emphasized the importance of evaluating the proposed method on additional datasets to validate its generalizability. The authors responded effectively by conducting experiments on diverse datasets, demonstrating consistent performance across varying scenarios.

Based on the reviewers’ positive feedback, the authors’ effective rebuttal, and the clear contributions of the paper, I recommend its acceptance.

**Additional Comments On Reviewer Discussion:**

The authors have provided detailed responses to all reviewers. Reviewers JKrr, N7nA, and HXs2 acknowledged that their concerns were effectively addressed in the rebuttal. The authors also clarified the concerns raised by Reviewer Wu7P, providing additional results and more detailed explanations, and Reviewer Wu7P has no further issues.

---

### Decision · Program_Chairs · 2025-01-22

Accept (Poster)